# Biocatalytic nanoparticles for the stabilization of degassed single electron transfer-living radical pickering emulsion polymerizations

Adrian Moreno [1✉] & Mika H. Sipponen [1✉]

Synthetic polymers are indispensable in many different applications, but there is a growing need for green processes and natural surfactants for emulsion polymerization. The use of solid particles to stabilize Pickering emulsions is a particularly attractive avenue, but oxygen sensitivity has remained a formidable challenge in controlled polymerization reactions. Here we show that lignin nanoparticles (LNPs) coated with chitosan and glucose oxidase (GOx) enable efficient stabilization of Pickering emulsion and in situ enzymatic degassing of single electron transfer-living radical polymerization (SET-LRP) without extraneous hydrogen peroxide scavengers. The resulting latex dispersions can be purified by aqueous extraction or used to obtain polymer nanocomposites containing uniformly dispersed LNPs. The polymers exhibit high chain-end fidelity that allows for production of a series of well-defined block copolymers as a viable route to more complex architectures.

[1] Department of Materials and Environmental Chemistry, Stockholm University, Svante Arrhenius väg 16C, SE-106 91 Stockholm, Sweden.
✉email: adrian.morenoguerra@mmk.su.se; mika.sipponen@mmk.su.se

Reversible deactivation radical polymerization (RDRP) techniques have emerged as one of the main tools for the efficient and precise synthesis of macromolecules with sophisticated and well-defined architectures for a wide range of macromolecular engineering applications[1–6]. Atom transfer radical polymerization (ATRP), reversible addition-fragmentation (RAFT) polymerization and single electron transfer-living radical polymerization (SET-LRP) offer high versatility to produce polymers, and the ability to minimize the termination events, while favoring equal propagation of polymeric chains during the polymerization process[7–9].

One of the main challenges of RDRP techniques arises from the presence of oxygen that is a well-known and undesired radical scavenger[10], which forces the application of degassing protocols or the introduction of reducing agents (e.g., hydrazine)[11–15], thus hampering the scale-up and transition from academia to industry of these controlled polymerization processes. To overcome this issue, notable works from the groups of Stevens[16–18] and Matyjaszwski[19–21], among others[22–25] reported the use of oxidoreductase enzymes to deplete oxygen, thus giving rise to enzyme-degassed RDRP techniques. Glucose oxidase (GOx, EC 1.1.3.4) is by far the most common enzyme applied due to its commercial availability and high efficiency to consume oxygen from the reaction mixture, allowing efficient polymerization processes in open-air systems[16,17]. So far, most of these systems have been applied in RAFT and ATRP of water-soluble monomers in aqueous media to retain the enzyme activity[20–24]. The polymerization of hydrophobic monomers remains scarcely explored, and is hitherto limited to ATRP-mediated miniemulsion processes using soluble ionic surfactants such as sodium dodecyl sulfate (SDS)[21], which could disrupt the activity of the enzyme at longer reactions times[26]. Indeed, some of these previously reported systems require the addition of sacrificial substrates (e.g., sodium pyruvate) to act as scavengers to consume the hydrogen peroxide generated during the GOx-catalyzed degassing step and thereby avoid giving rise to detrimental Fenton-like redox processes[19,21].

Taking into account the growing interest towards polymerization in dispersed media in both academia and industry, one of the current challenges is to develop surfactant/emulsifier systems that stabilize emulsions of hydrophobic vinyl monomers for RDRP, while simultaneously enabling enzymatic deoxygenation of water and scavenging of the hydrogen peroxide formed during the degassing process. Pickering emulsion polymerizations stabilized by organic or inorganic particles have gained a great attention over conventional emulsion polymerization processes due to a more-stable emulsification behavior, and the ability to produce composites with improved properties[27,28]. These systems have been scarcely applied in RDRP techniques such as ATRP or RAFT to generate hybrid latexes with a reasonable extent of control[29,30]. In addition, the literature is lacking reports on the development of SET-LRP-mediated Pickering emulsion processes in general and enzyme-degassed RDRP-mediated Pickering emulsion processes in particular.

Here, we report a multifunctional Pickering emulsion stabilizer that enables efficient SET-LRP of water-immiscible vinyl monomers without extra degassing steps and at a broad temperature range. We use lignin nanoparticles (LNPs) as a low-cost and green scaffold to adsorb cationic chitosan polymer, and use this system as a carrier for GOx via adsorption. The non-covalent synthesis of these cationic hybrid particles was encouraged by their advantageous performance for the stabilization of Pickering emulsions[31,32], and enzyme immobilization[33–35]. We also rationalized their potential ability to self-scavenge H$_2$O$_2$ during the polymerization, which was expected to be beneficial for the activity of GOx and control the polymerization without the use of any extraneous reducing agents.

In addition to investigating the ability of the hybrid particles to scavenge the in situ generated hydrogen peroxide and produce well-defined polymers, we also explore the obtained latex dispersions for the preparation of lignin-reinforced composites. We demonstrate that a simple melt processing of the latex dispersions gives composites with uniform dispersion of antioxidant and UV-protective LNPs in the polymeric matrixes, resolving in this way one of the main challenges in the preparation of synthetic polymer composites with bio-based particulate fillers.

## Results

**Fabrication of biocatalytic hybrid particles.** Our approach to fabricate multifunctional Pickering emulsion polymerization stabilizers started with production of biocatalyst-loaded LNPs (GOx-chi-LNPs) via a two-step adsorption immobilization strategy (Fig. 1).

First, LNPs (ζ-potential −40 mV) were produced by solvent-exchange methodology from previously characterized pine kraft lignin (Supplementary Table 1)[33]. The resulting LNPs (diameter 97 nm) were used to adsorb chitosan (10 wt% relative to LNPs) as a cationic polyelectrolyte. As previously reported[32], we obtained colloidally stable cationic chitosan-coated LNPs (chi-LNPs) (ζ-potential +32 mV and diameter 190 nm). In the present study, we aimed to develop a material-efficient process and in order to simplify the LNPs production process we did not use dialysis as a purification step but instead evaporation for the solvent removal. Therefore, the larger than expected increase in the particle diameter after the chitosan adsorption could be related to the co-precipitation of chitosan and water-soluble lignin compounds bearing carboxylic acid groups on the LNPs surface[32]. Afterwards, chi-LNPs were used to adsorb GOx (10 mg g⁻¹ chi-LNPs) to yield the enzyme-immobilized LNPs (GOx-chi-LNPs) (Supplementary Figs. 1 and 2). The increase in size and surface charge as determined by dynamic light scattering (DLS) analysis of the resulting particles after the two-step adsorption procedure (215 nm and +41 mV) verified an effective electrostatic interaction between GOx and chi-LNPs (Fig. 2a, b and Supplementary Table 2).

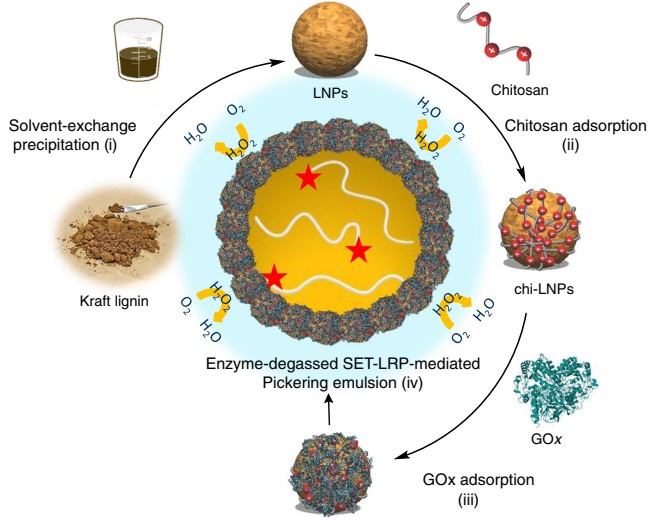

**Fig. 1 General process scheme for the preparation of GOx-chi-LNPs via a two-step adsorption immobilization process and application in enzyme-degassed SET-LRP-mediated Pickering emulsion process.** (i) Preparation of LNPs via solvent-exchange precipitation. (ii) adsorption of chitosan on LNPs to yield chi-LNPs. (iii) Adsorption of GOx on chi-LNPs to yield GOx-chi-LNPs. (iv) Application of GOx-chi-LNPs as biocatalytic degassing stabilizers for SET-LRP in Pickering emulsion media.

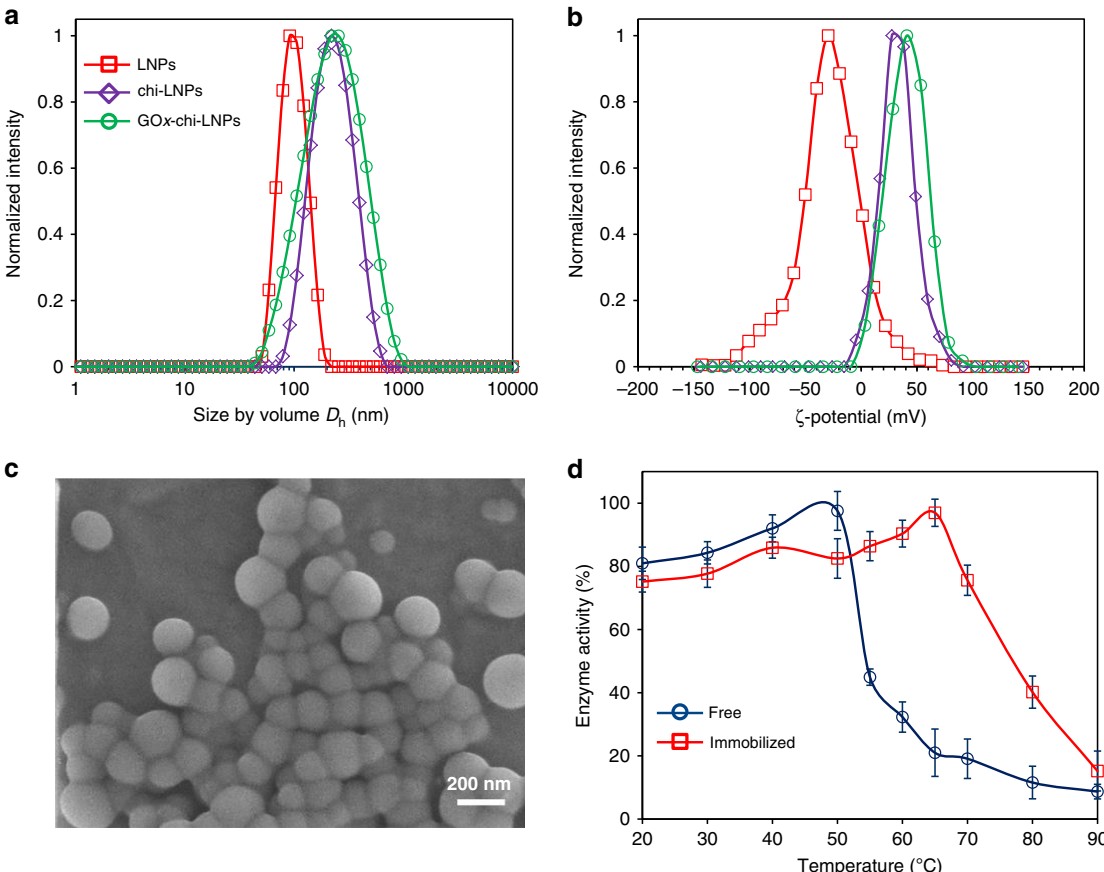

**Fig. 2 Characterization of GOx-chi-LNPs. a** Hydrodynamic diameter and **b** ζ-potential distributions of lignin nanoparticles employed in this work. (red squares) LNPs, (purple diamonds) chi-LNPs and (green circles) GOx-chi-LNPs. The mean values ± SD ($n = 3$) of the particle diameter and ζ-potential results in **a**, **b** are given in Supplementary Table 2. **c** SEM image of GOx-chi-LNPs. **d** Thermal stability of (blue circles) free and (red squares) immobilized GOx. Mean values ± SD ($n = 3$) are shown.

Scanning electron microscopy (SEM) images confirmed the formation of spherical and uniformly shaped biocatalyst-loaded particles (GOx-chi-LNPs) that exhibited a high tendency to agglomerate upon drying (Fig. 2c). A protein mass balance based on Bradford protein assay (BSA standards), showed that 88% of GOx adsorbed on chi-LNPs, whereas the remaining 12% was present in the solution or adsorbed on the smallest particles that did not sediment during the centrifugation step used in the sample preparation for the protein assay.

As polymerization reactions are carried out in a broad range of temperatures, we first studied thermal stability of the immobilized GOx in comparison to the free enzyme. It turned out that the immobilized GOx remained active until 80 °C, with a temperature optimum at 65 °C compared to ~50 °C of the free enzyme (Fig. 2d). One possible explanation for this increased stability is the ability of the chi-LNPs to scavenge hydrogen peroxide produced by GOx (Fig. 1, and vide infra for more details). Comparison before and after the immobilization step of enzyme activity also confirmed a high retention of activity, which could be associated to a low kinetic constraint from the embedment of the enzyme on the chitosan hydrogel layer over the LNPs (Supplementary Fig. 3 and Supplementary Table 3).

**Enzyme-degassed SET-LRP-mediated Pickering emulsion using GOx-chi-LNPs as emulsifiers.** With the biocatalyst-loaded particles available as a colloidally stable dispersion, we assessed whether or not the biocatalytic emulsifier system is suitable for

conducting SET-LRP reactions. All the polymerizations were conducted in aqueous buffer solutions (pH = 6) at 50 °C in Pickering emulsions produced by mixing the monomer, ligand and initiator (oil phase) with an aqueous dispersion containing GOx-chi-LNPs (20 g per L of monomer) and glucose (0.1 M) by ultra-sonication (See Supplementary Methods section and Supplementary Table 4 for more details about the polymerization conditions). The emulsion droplets were efficiently covered by GOx-chi-LNPs (Supplementary Fig. 4 and Supplementary Table 5) regardless of the monomer employed, and polymerization was initiated by adding a small volume of aqueous Cu(0) powder dispersion into the Pickering emulsion. Nanosized Cu(0) particles (40–60 nm) were used to ensure a high active surface area and provide a fast initiation step, crucial to the propagation step[36]. The polymerization reactions were allowed to run in either open or closed vials. It is important to note that in our system, activation, disproportionation and deactivation steps take place in different compartments via partitioning of the ligand and copper active species, as has been recently postulated for "programmed" biphasic SET-LRP systems[37–39]. In this sense, the activation step proceeds in the organic phase (oil phase) via outer sphere single electron transfer process (OSET), where Cu(0) acts as an electron donor to promote the heterolytic cleavage of the alkyl halide bond to generate the active radicals[40]. Afterwards, the generated Cu(I)-ligand complex formed during the activation step migrate to the aqueous phase where the crucial disproportionation step takes place, giving rise to the Cu(II)-ligand complex and in situ nascent Cu(0). In this context, the deactivation step is postulated to take place in the

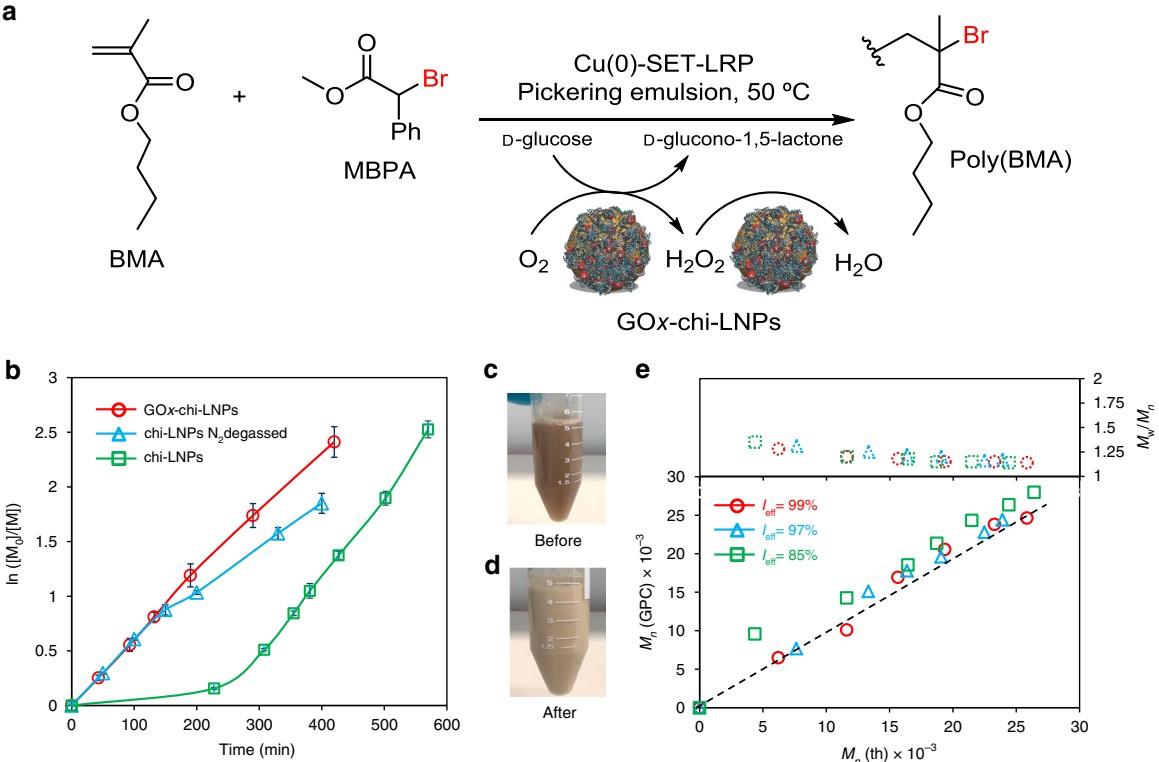

**Fig. 3 Evaluation of the enzyme-degassed controlled radical polymerization of BMA using GOx-chi-LNPs as stabilizers. a** Scheme of enzymatic-degassed SET-LRP-mediated Pickering emulsion of BMA. **b** Kinetic plots and **e** evolution of experimental $M_n$ (GPC) and $M_n/M_w$ vs theoretical $M_n$(th) for the SET-LRP-mediated Pickering emulsion of BMA using GOx-chi-LNPs (red circles), chi-LNPs (blue triangles) in the presence of oxygen, and chi-LNPs (green squares) in $N_2$ degassed emulsion. Mean values ± SD ($n = 2$) are shown in **b**, whereas all data points are shown in **e**. **c, d** Digital images of BMA-based Pickering emulsions before **c** and after **d** SET-LRP process using GOx-chi-LNPs as stabilizers. Reaction conditions: $[BMA]_0/[MBPA]_0/[Me_6-TREN]_0/[Cu(0)]_0 = 200/1/0.2/0.3$. [Lignin nanoparticles] = 2.25 wt% relative to BMA.

interphase between the aqueous and oil phases, via a reverse OSET process mediated by Cu(II)-ligand complex[37]. Therefore, the surface area of the polymeric microdroplets and an efficient partition of the ligand between aqueous and organic (oil) phases are crucial aspects in our system to provide an efficient deactivation step and therefore provide control over the polymerization.

The feasibility of GOx-chi-LNPs to provide an oxygen-tolerant and controlled polymerization process was evaluated by the polymerization of butyl methacrylate (BMA) as a model monomer in closed vials to avoid the loss of volatile BMA from the reaction mixture (Fig. 3a). A kinetic polymerization study revealed a pseudo-first-order polymerization kinetics and good linearity ($R^2 = 0.98$) between $\ln([M_0]/[M])$ vs time (Fig. 3b, red circles), indicating an equal and constant propagation of the growing polymeric chains, which essentially confirms a lower frequency of the termination events (i.e., recombination of growing polymeric chains). These results, together with a high retention of chain-end functionality, and the ability to reinitiate the polymeric chains (vide infra), prove unequivocally the living radical polymerization behavior for our process. In addition, any appreciable induction period could not be observed, which confirms an efficient degassing process catalyzed from GOx-chi-LNPs. As a visual observation, it was noted that the appearance of the emulsion became opaquer as a result of the polymerization (Fig. 3c, d). The high tolerance to air atmosphere was additionally assed by temporally opening and closing the reaction vial in 20 minutes intervals during 4 hours, which essentially confirmed that exposure to air had no effect on the polymerization rate or extent of control even at the most oxygen-exposed periods (open vial times) (Supplementary Fig. 5). For comparison, chi-LNPs

without GOx were used by applying nitrogen gas purging as a degassing methodology (Fig. 3b, blue triangles) and without any degassing protocol (Fig. 3b, green squares). It can be very clearly observed that the system of chi-LNPs without degassing measures showed an induction period of >3 h before the polymerization began to proceed, which confirms previous findings on the inhibition of the reaction by dissolved oxygen[11,12]. In the case of chi-LNPs, applying $N_2$ bubbling for degassing, a similar kinetic profile than that observed for GOx-chi-LNPs could be obtained, albeit at a slightly slower polymerization rate, which could be associated to the inevitable oxygen contamination, resulting from the sampling process despite applying concurrent purging with $N_2$.

These results not only validate our original hypothesis and confirm GOx-chi-LNPs as efficient degassing stabilizers for Pickering emulsions polymerizations, but also demonstrate that GOx-chi-LNPs can provide even a faster polymerization rate than chi-LNPs after applying nitrogen purging, which could be of potential interest for industrial applications. Moreover, molecular weight analysis at different polymerization times by gel permeation chromatography (GPC) revealed a linear increase of molar mass with a good agreement with the theoretical molecular weight values, leading to well-defined polymers ($M_w/M_n = 1.30-1.15$) during the whole polymerization process for GOx-chi-LNPs and chi-LNPs ($N_2$ degassed) (Fig. 3e, red circles, blue triangles, and Supplementary Fig. 6). These results confirm the potential of LNPs as functional emulsifiers for SET-LRP. However, it is important to note that in the case of chi-LNPs without applying any degassing procedure, a clear deviation from the $M_n$ (GPC) values in comparison to the theoretical ones

$M_n$ (th), can be appreciated, especially at the early stages of the reaction (Fig. 3e, green squares). This fact, together with a decrease in the initiator efficiency ($I_{eff}$, Fig. 3e), is associated to the consumption of the dissolved oxygen at the beginning of the polymerization process by the growing oligomeric radicals.

**Self-scavenger ability of GOx-chi-LNPs towards hydrogen peroxide.** Inactivation by hydrogen peroxide of enzymes, in general, and GOx, in particular, is a problem that has been alleviated by additives such as the electron acceptor system benzoquinone–hydroquinone[41]. Moreover, as already mentioned, $H_2O_2$ may initiate Fenton-like oxidation processes and promote the generation of new chains, leading to uncontrolled growth of the polymeric chains, which is detrimental to final molecular weight distribution[19]. It is thus important to note that our system was able to generate well-defined polymers without the need of any extraneous reducing agents (e.g., sodium pyruvate) to eliminate the $H_2O_2$ formed during the degassing step. This fact could be attributed to the inherent nature of both chitosan and lignin to consume $H_2O_2$ in situ[42,43]. To test this hypothesis, we evaluated the ability of GOx-chi-LNPs to consume $H_2O_2$. First, $H_2O_2$ was dissolved in an aqueous solution (pH = 6) and mixed with horseradish peroxidase (HRP, EC 1.11.1.7) in opened vials with and without LNPs or GOx-chi-LNPs, and then $o$-dianisidine was added as a substrate for HRP (Supplementary Fig. 7a). In the absence of LNPs or GOx-chi-LNPs, a significant increase in absorption at 500 nm was observed by UV–vis spectroscopy owing to the formation of the oxidized dimer of $o$-dianisidine ($\varepsilon_{[500\,nm]} = 7.5$ mM in water) catalyzed by HRP in the presence of $H_2O_2$ (Supplementary Fig. 7b). In contrast, when the reaction was performed in the presence of LNPs, a noteworthy decrease in the UV absorption of oxidized $o$-dianisidine indicated an effective consumption of $H_2O_2$ by LNPs (Supplementary Fig. 7c). In addition, it was interesting to observe that when GOx-chi-LNPs were added to the reaction mixture, only trace levels of the oxidized product could be determined, suggesting a cumulative combined effect of lignin and chitosan to scavenge the $H_2O_2$ present in the system (Supplementary Fig. 7d, e). In addition, to mimic the polymerization conditions, the same amount of BMA, glucose and GOx-chi-LNPs were dissolved in an aqueous solution (pH = 6) and exposed to air. Then, $o$-dianisidine and HRP solution were added, and the analysis of the resulting solution confirmed that the mere presence of GOx-chi-LNPs suppressed the formation of $H_2O_2$ in the reaction media (Supplementary Fig. 8). These results confirm the scavenger ability of our system toward $H_2O_2$, which is beneficial not only for the polymerization process but also to protect the enzyme against the oxidative denaturation[44], which could also explain the high retention of enzyme activity that the immobilized GOx exhibited at high temperatures (Fig. 2d).

**Applicability to other monomers and synthesis of well-defined block copolymers.** Having demonstrated the robust nature of GOx-chi-LNPs to act as functional surfactants to obtain well-defined hydrophobic polymers from oxygen-tolerant SET-LRP, we explored versatility of the system by targeting the polymerization of different vinyl monomer of relevant industrial interest. For this purpose, we conducted polymerizations under identical degree of polymerization (DP = 200) to those applied for BMA for two additional hydrophobic monomers, namely methyl acrylate (MA) and styrene (S). In both cases, near to quantitative monomer conversion (90%) was achieved, and molecular weight analysis of the resulting polymers revealed symmetrical monomodal peaks ($M_w/M_n = 1.16$ for MA and 1.26 for S) with a close agreement between the experimental and theoretical molecular weights values (Fig. 4a), proving the versatility of this system to polymerize, in a controlled manner, different families of hydrophobic vinyl

monomers in oil-in-water Pickering emulsions. We also decided to push the limits of our system by targeting different degrees of polymerization ($DP_n$), ranging from 50 to 500 for BMA. The results are summarized in Supplementary Table 6. In all cases, well-defined polymers with narrow dispersities ($M_w/M_n = 1.29-1.16$) and experimental molecular weight values close to the theoretical ones could be obtained, indicating a high versatility of the system to deliver well-defined polymers in a wide range of molecular weights (Fig. 4b and Supplementary Fig. 9).

The high chain-end fidelity of the obtained polymers, a critical parameter to synthetize more complex architectures (e.g., block and multiblock copolymers)[45–47] was also assed for our system. Low molecular weight PMA (M/I = 50) using GOx-chi-LNPs as emulsifiers was synthetized, isolated and analyzed by $^1$H NMR spectroscopy, which revealed virtually complete chain-end functionality (96%) based on the retention of the bromine chain-ends (Supplementary Fig. 10). For comparative purpose, PMA (M/I = 50) using chi-LNPs and without the application of any deoxygenating protocols was also synthetized. In this case, the chain-end analysis of the sample revealed a lower functionality (85%) (Supplementary Fig. 11) in comparison to the above mentioned, which essentially confirms the occurrence to some extent of bimolecular termination events (i.e., by the coupling of growing oligomeric radicals with dissolved oxygen). In addition, chain extension experiments using GOx-chi-LNPs by stopping the reaction, purification of the macromonomer through precipitation and re-dispersion to continue the polymerization of a second block also confirmed a nearly perfect chain-end functionality regardless of the monomer employed (Fig. 4c and Supplementary Fig. 12). The successful formation of well-defined block copolymers was confirmed by a clear shift in the GPC curves toward high molecular weight values without noticeable shoulders or tailing (Fig. 4c and Supplementary Fig. 12), indicating the absence of unreactive polymeric chains produced in recombination processes of growing polymeric chains (e.g., bimolecular termination processes). Overall, these results not only validate GOx-chi-LNPs as an efficient stabilizers for more sophisticated controlled polymerization techniques in the presence of oxygen, but also revealed their high versatility to stabilize and enable SET-LRP of different vinyl monomers leading to well-defined polymers as building units for more complex and sophisticated architectures.

**Analysis of latex dispersion and production of lignin-based composites.** Finally, the quality of the latex dispersions obtained after the SET-LRP process was also evaluated by SEM and optical light microscopy. Regardless of the monomer employed, the latex dispersions were in all cases composed of uniform spherical polymeric microbeads (Fig. 5 and Supplementary Fig. 14), with a similar size to that of the initial monomer droplets (compare Supplementary Tables 5 and 7). As a representative example, BMA latex dispersion was selected for a more-detailed analysis (Fig. 5). Lignin-coated poly(butyl methacrylate) (PBMA) microparticles could be obtained by isolation of the particles by a simple centrifugation/re-dispersion step. Based on the microscopic images, it can be clearly observed that the PBMA microspheres contained a thin amorphous layer (Fig. 5b, c), which demonstrates the high emulsifier capacity of the GOx-chi-LNPs system. Such a uniform distribution of the nanoparticles in the polymeric matrix inspired us to envision that the polymerization system could also provide an effective and direct method to prepare lignin-based composites with homogeneously dispersed lignin particles in the polymeric matrix. With this objective in mind, we evaluated the preparation of composites by simple melting at 160 °C of the PBMA latex dispersion stabilized with a

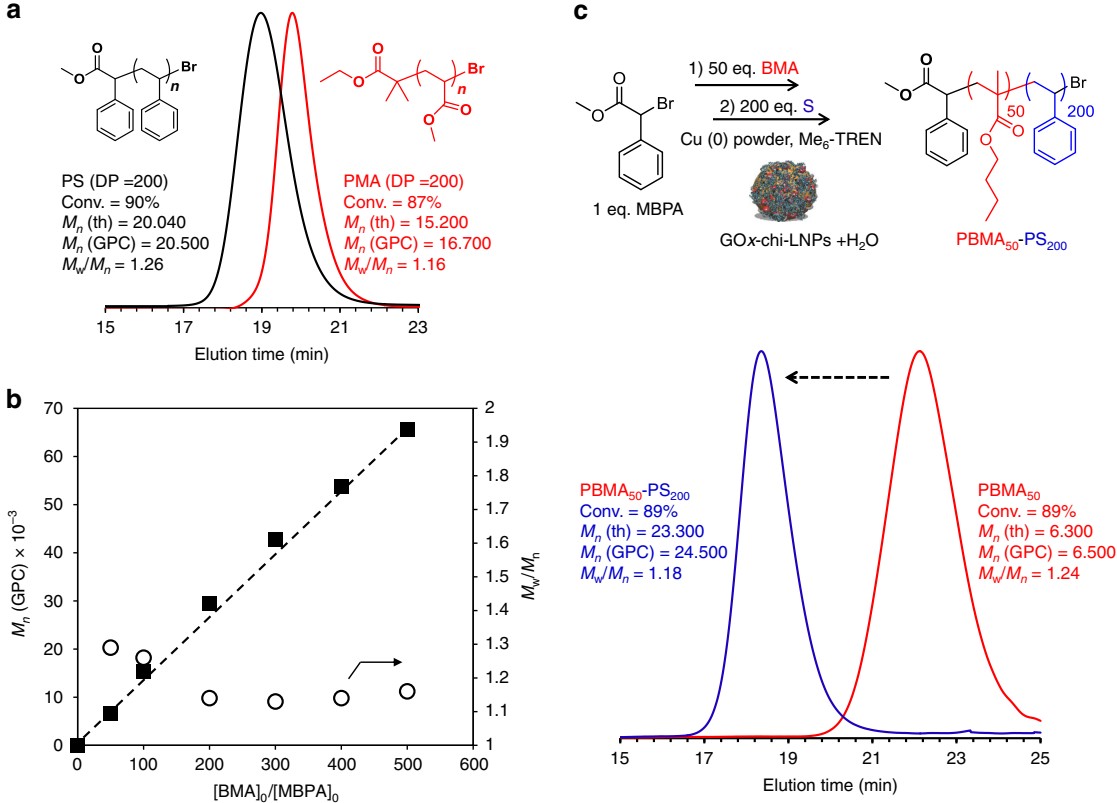

**Fig. 4 Expanding the monomer scope and synthesis of block copolymers. a** GPC curves of poly(methyl acrylate) (PMA) and polystyrene (PS) obtained by enzyme-degassed Pickering emulsion SET-LRP. Reaction conditions: $[\text{Monomer}]_0/[\text{MBPA or EBiB}]_0/[\text{Me}_6\text{-TREN}]_0/[\text{Cu(0)}]_0 = 200/1/0.2/0.3$. [Lignin nanoparticles] = 2.25 wt% relative to MA or S. **b** Dependence of experimental $M_n$ (GPC) and $M_w/M_n$ on the $[\text{BMA}]_0/[\text{MBPA}]_0$ ratio. The mean values ± SD ($n = 2$) of BMA conversion for the different DPs are given in Supplementary Fig. 13, whereas all data points are shown in **b**. **c** GPC analysis of block copolymerization via purification-reinitiation strategy of poly(butyl methacrylate) (PBMA) with styrene (S) to synthetize a PBMA-PS block copolymer. Reaction conditions for the PBMA macroinitiator: $[\text{BMA}]_0/[\text{MBPA}]_0/[\text{Me}_6\text{-TREN}]_0/[\text{Cu(0)}]_0 = 50/1/0.2/0.3$. [GOx-chi-LNPs] = 2.25 wt% relative to BMA.

higher content of GOx-chi-LNPs (6 wt% relative to BMA) (Supplementary Fig. 15a). SEM analysis of the surface and cross-sections of PBMA-GOx-chi-LNPs film composites (Supplementary Fig. 15b, c) confirmed the uniform distribution of GOx-chi-LNPs without agglomeration within the PBMA matrix, which indicates a high and effective interaction of the particles even within such hydrophobic polymeric matrixes. These results make it possible to form lignin-based polymer composites in a simple way, avoiding chemical functionalization of lignin to improve the dispersability within the polymeric matrix. It is also interesting to note that the unpurified polymer composite contained <0.2% of copper when analyzed by SEM-EDX, while showing 2.7 atom% of bromine linked to the active chain-ends (Supplementary Fig. 16). However, it is important to state that the presence of copper traces in the final composite could be expected, as lignin has affinity to adsorb copper (15 mg/g)[48]. On the other hand, the GOx-chitosan-lignin hybrid particles (GOx-chi-LNPs) displayed a positive net charge, which should be repulsive against the adsorption of copper cations. Last but not least, it is also important to note that highly purified PBMA microparticles without the presence of lignin could also be obtained by simple aqueous extraction of the latex dispersions under alkaline conditions (Fig. 5d–f).

## Discussion
We have developed a multifunctional particulate emulsifier based on enzyme-immobilized LNPs (GOx-chi-LNPs), and demonstrated their application in oxygen-tolerant SET-LRP in Pickering emulsions. This constitutes the first report on enzyme-degassed SET-LRP reactions, and enzyme-degassed RDRP-mediated Pickering emulsion processes in particular. The rational design of our GOx-chi-LNPs was not limited to enabling the synthesis of hydrophobic polymers in a controlled manner under air atmosphere, but also to enhance the enzyme activity in a wide range of temperatures, together with a self-scavenger ability towards $H_2O_2$ during the polymerizations without adding external additives. In addition, the potential of this approach has been highlighted by the efficient preparation of lignin-based composites, with an expected favorable carbon footprint by simple melt processing. Finally, we also hold a view that this system can be easily transferred to other polymerization techniques such as ATRP, RAFT, or FRP and thus holds potential for opening new avenues in the preparation of precision polymers and lignin-polymeric composites under green conditions.

## Methods
**Preparation of colloidal lignin particles (LNPs, chi-LNPs, and GOx-chi-LNPs).**
LNPs used in this work were produced following the same procedure described earlier with a few modifications[32]. In brief, kraft lignin was dissolved in acetone/water mixture (mass ratio 3:1), insoluble impurities were removed by filtration, and LNPs produced by rapid pouring of deionized water to lignin solution followed by rotary evaporation of acetone. The final aqueous dispersion of LNPs (0.4 wt%) was obtained after filtration with a lignin mass yield of 89%. The chitosan-coated LNPs (chi-LNPs) were prepared by adding the dispersion of LNPs under vigorous stirring into a 0.1 wt% chitosan solution. The ratio of chitosan to LNPs was 100 mg g⁻¹. GOx-chi-LNPs were produced by the addition of a GOx solution (sodium acetate buffer, 0.1 M; pH 5) to chi-LNPs dispersion under orbital shaking at room temperature for 2 h. GOx-chi-LNPs were recovered by three centrifugation/re-dispersion cycles (3000 rpm, 10 min), replacing each decanted supernatant with sodium acetate buffer (0.1 M; pH 5). The ratio of GOx to chi-CLPs was 10 mg g⁻¹.

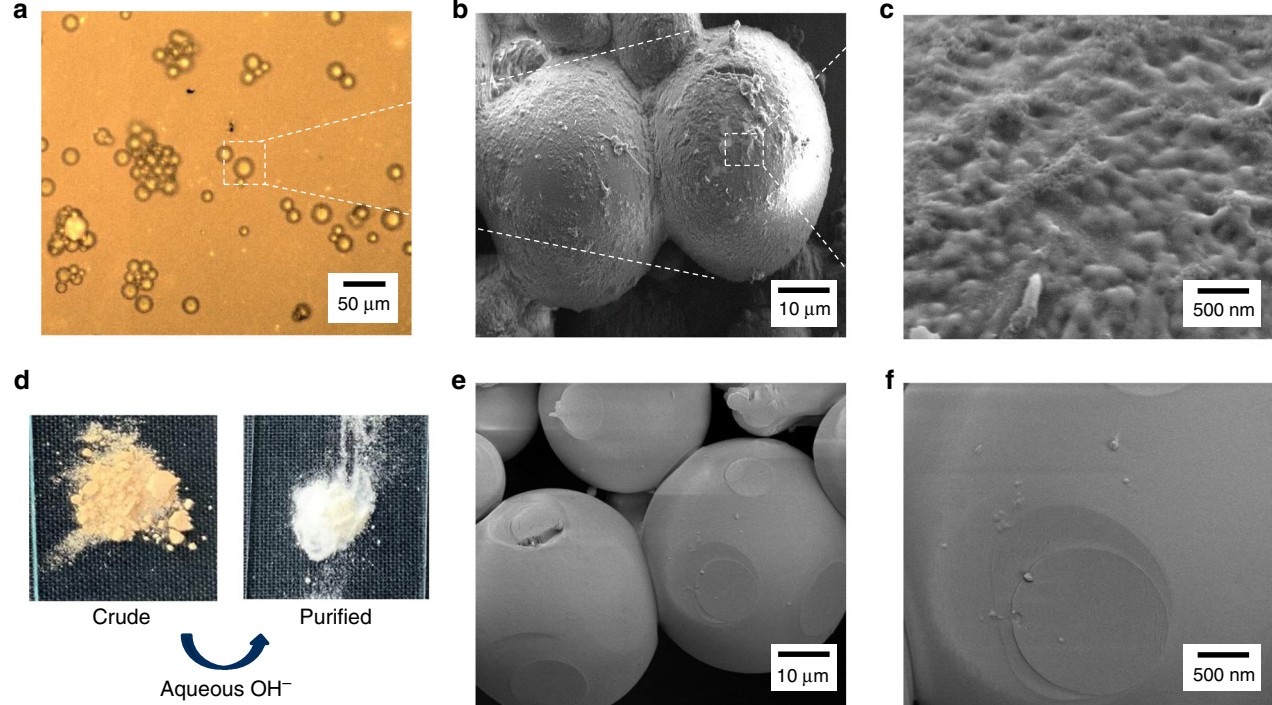

**Fig. 5 Characterization of PBMA-GOx-chi-LNPs latex dispersions. a** Optical microscopic and **b**, **c** SEM images of the lignin-coated PBMA microbeads after SET-LRP process. **d** Digital images of PBMA before and after the purification step using basic aqueous solution (NaOH, 0.1 M). **e**, **f** SEM images of bare PBMA microbeads obtained after the purification from the lignin-coated analogous.

**Preparation of GOx-chi-LNPs stabilized BMA-in-water Pickering emulsions.**
All the emulsions were prepared by gradually adding BMA monomer to a water dispersion of GOx-chi-LNPs. The final fraction of oil/water was fixed at 20/80 v/v and the total volume of the emulsion was 10 mL. The final concentration of GOx-chi-LNPs was fixed to 20 g of GOx-chi-LNPs per L (2.25 wt%) of BMA. The emulsification was performed by sonication for 120 s with a BioBlock Vibra-Cell equipped with an ultrasonic tip with cooling in an ice bath (10 s on and 5 s off at 40% of amplitude power).

**Enzyme-degassed SET-LRP-mediated Pickering emulsion process.** This procedure is representative for all the polymerization conducted herein. The enzyme-degassed Pickering emulsion SET-LRP of BMA using GOx-chi-LNPs as emulsifier was carried out under the following conditions: $[BMA]_0/[MBPA]_0/[Me_6-TREN]_0/[Cu(0)]_0 = 200/1/0.2/0.3$. [GOx-chi-LNPs] = 2.25 wt% relative to BMA, is described as a representative procedure. A stock solution of BMA (2 mL, 0.012 mol), MBPA (11 μL, 0.062 mmol) and $Me_6$-TREN (3.6 μL, 0.012 mmol) was prepared. GOx-chi-LNPs (40 mg, 2.25 wt% to BMA) were dispersed in water (7.5 mL) containing glucose (200 mg, 0.14 M). Then, BMA-in-water Pickering emulsion were prepared by ultrasonication of organic phase (oil phase) in water phase as described above. The Pickering emulsion was transferred to a vial and placed in a thermostatic oil-bath at 50 °C. The introduction of an aliquot of Cu(0) powder dispersed in water (1.20 mg, 0.5 mL) started the SET-LRP process ($t = 0$). In general, the reactions were allowed to proceed during 12 h, and in the case of kinetic experiments, samples were withdrawn periodically to follow the monomer conversion by gravimetric analysis. The resulting lignin-polymeric microparticles were purified by three centrifugation/re-dispersion cycles, replacing each decanted supernatant with aqueous basic solution (NaOH, 0.1 M), followed by drying overnight of the purified polymer samples at 45 °C for 12 h prior to the GPC analysis. To purify the polymeric particles without removing lignin from microparticle surface, deionized water was used instead of basic washing solution.

## Data availability

The DLS, zeta potential, ¹H NMR, GPC traces, enzyme activity and stability, and source data that support the findings of this study are available in Zenodo repository at https://doi.org/10.5281/zenodo.4054808. Other data are available within the article and Supplementary Information or from the authors upon reasonable request. Source data are provided with this paper.

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

## Acknowledgements
The authors acknowledge the Department of Materials and Environmental Chemistry (MMK) for financing this work through the start-up grant awarded to MHS. The authors also acknowledge Dr. Kjell Jansson for helpful discussions, and Professor Lennart Bergström and Dr. Claudia Möckel for giving access to ultrasonication and GPC instrumentation, respectively.

## Author contributions
A.M. and M.H.S. conceived the idea and designed the experiments. A.M. performed the experiments and analyzed the data with inputs from M.H.S. A.M. and M.H.S. co-wrote the manuscript.

## Funding

## Competing interests
The authors declare no competing interests.
