## [Peer Review File · Nature Communications]

REVIEWER COMMENTS

Reviewer #1 (Remarks to the Author):

Review of the manuscript titled: "Biocatalytic Hybrid Nanoparticles for Degassing and Stabilization of Single Electron Transfer-Living Radical Polymerization in Pickering emulsions" submitted to Nature Communications

Moreno and coauthors developed a strategy to polymerized hydrophobic vinyl polymers without the need of any degassing procedure and using water as the main solvent. For this the author stabilized the monomer droplets using a ligning nanoparticles to which a cationic polymer is bound which trapped GOx. The enzyme scavenge hydrogen peroxide allowing to proceed with the polymerization in a control manner. This enzyme has already been utilized as "deoxygenation agent" (properly cited by the authors) however, the novelty of this work lays in addressing a key challenge, a readily translatable way to perform controlled radical polymerization without degassing and using water as the main solvent. The authors carefully designed the experiments to prove their hypothesis demonstrating that their system could afford narrowly disperse polymers that were living (semitelechelic). All the conclusion are well supported by the data. This is certainly an interesting and very well executed work, which should be published after minor revisions, noted below.

Minor queries/comments:

Introduction:

The authors comment that the chitosan is favorable for stabilization but it is not clear whether they refer to the emulsion or to the enzyme. How does the charge affect the activity of the enzyme?

Results:

1) Can the authors comment about the increment in thickness upon chitosan adsorption? It seems rather larger for a layer-by-layer deposition.

2) What is the partition of the ligand between the oil and water phase? The author should discuss the need of some Me6TREN in the water phase to stabilized the Cu²⁺

3) "linearity between indicating an equal propagation of the growing polymeric chains, the principal feature of a living polymerization" The principal feature of a living polymerization is the ability to re-initiate and chain extend. The linear plot of $\ln(M/M_0)$ vs t is a feature of control in the degree of polymerization. The authors should rephrase their sentence.

4) Figure 3 reports the comparison between GOx-chit-CLPS and chi-CLPs. It is reassuring to see that the enzyme was capable of degassing and protecting the system (compared to N₂ degassing) and prevented any induction period. I am confused however, why in the plot of Mn (GPC) vs Mn (theo) all the kinetics lay on the same curve. I would expect that the presence of oxygen would cause termination and a concomitantly lower initiation efficiency. The manuscript would benefit from the calculation of the initiation efficiency and chain-end functionality in this comparison.

5) What do the author mean with "quasi real AB block copolymers"?

6) The direct use of the stabilized polymer emulsion is interesting for many applications. Have the authors check if there is any remaining Cu in the polymer phase? And how much is it?

In summary, this is an excellent contribution of Moreno and Sipponen, which allows for the polymerization of hydrophobic polymers in water without the need of any degassing and even protect the polymerization from oxygen coming into contact during the polymerization. This is an excellent way to achieve polymers with controlled distribution of molecular weight and high chain end functionality in very facile manner. I am convinced that the findings presented here will be of great interest for a broad readership and I recommend the acceptance of this manuscript.

César Rodríguez-Emmenegger

Reviewer #2 (Remarks to the Author):

This is an excellent manuscript that will impact the field of living radical polymerization and therefore I recommend publication after minor revisions. The following minor revisions must be considered by the authors: (1) In reference 10 the name of the first author must be corrected to S. Fleischmann and of the third author to V. Percec. The following additional citations on the topic of reference 10 could be cited: JPS Polym Chem 2010, 48, 2243; JPS Polym Chem 2013, 51, 3110 and JPS Polym Chem 2011, 49, 4756. (2) In Figure 4a,b and S5b and S6a,b,c replace Mn/Mw with Mw/Mn. After these minor revisions this excellent manuscript is recommended for publication.

Manuscript number: NCOMMS-20-29651

Manuscript type: Research article

Title: Biocatalytic Hybrid Nanoparticles for Degassing and Stabilization of Single Electron Transfer-Living Radical Polymerization in Pickering emulsions

Authors: Adrian Moreno, Mika H. Sipponen

Correspondence to: mika.sipponen@mmk.su.se

Responses to Reviewer's comments:

We would like to thank the expert reviewers for their critical comments on our manuscript. We have taken into account all the insightful comments of the reviewers as well as editorial instructions and revised the manuscript accordingly. Our point-by-point responses to the reviewers are included in blue font after each reviewer comment that is available in black. The changes made to the manuscript are shown in red color. A clear revised version of the manuscript and a red marked version with all changes clearly indicated are included in this submission.

REVIEWER COMMENTS

Reviewer #1 (Remarks to the Author):

Review of the manuscript titled: "Biocatalytic Hybrid Nanoparticles for Degassing and Stabilization of Single Electron Transfer-Living Radical Polymerization in Pickering emulsions" submitted to Nature Communications

Moreno and coauthors developed a strategy to polymerized hydrophobic vinyl polymers without the need of any degassing procedure and using water as the main solvent. For this the author stabilized the monomer droplets using a lignin nanoparticles to which a cationic polymer is bound which trapped GOx. The enzyme scavenge hydrogen peroxide allowing to proceed with the polymerization in a control manner. This enzyme has already been utilized as "deoxygenation agent" (properly cited by the authors) however, the novelty of this work lays in addressing a key challenge, a readily translatable way to perform controlled radical polymerization without degassing and using water as the main solvent. The authors carefully designed the experiments to prove their hypothesis demonstrating that their system could afford narrowly disperse polymers that were living (semitelechelic). All the conclusion are well supported by the data. This is certainly an interesting and very well executed work, which should be published after minor revisions, noted below.

Answer: Thank you very much for reviewing our manuscript, the positive comments about our work and recommending it for publication after minor revisions.

Minor queries/comments:

Introduction:

The authors comment that the chitosan is favorable for stabilization but it is not clear whether they refer to the emulsion or to the enzyme. How does the charge affect the activity of the enzyme?

Answer: We thank the reviewer for the possibility to clarify this aspect. We selected chitosan as the cationic polymer because we expected it to have a beneficial impact on both enzyme immobilization and emulsion stability when adsorbed on lignin nanoparticles (ref. 31-33). In addition to the need for stabilization of Pickering emulsions of hydrophobic vinyl monomers, we reasoned that the system might have a potential self-scavenger ability towards H_2O_2 . After the enzyme immobilization step, we only observed a slightly decrease in enzyme activity. We assume that this fact could be more related to a less favorable 3D shape-distribution of the enzyme when embedded into the viscous chitosan hydrogel layer over LNPs than to the effect of the charge. As demonstrated, the in situ scavenging of H_2O_2 ultimately turned out crucial for the enzyme stability and control over the polymerization. We also took into consideration that the chi-LNPs could act as an efficient carrier for GOx via electrostatic interactions promoted by a high surface area, and that the final biocatalytic particles GOx-chi-LNPs remained cationic and thus beneficial for emulsion stabilization. We have stressed this in the introduction of the revised manuscript as follows: **We use lignin nanoparticles (LNPs) as a low-cost and green scaffold to adsorb cationic chitosan polymer and use this system as a carrier for GOx via adsorption. The non-covalent synthesis of these cationic hybrid particles was encouraged by their advantageous performance for the stabilization of Pickering emulsions.^{[31][32]} We also rationalized their potential ability to self-scavenge H_2O_2 during the polymerization, which was expected to be beneficial for the activity of GOx and control the polymerization without the use any extraneous reducing agents.^[33-35]** Please, see page 1, left column lines 1-6 from top of the red market manuscript)

Results:

1) Can the authors comment about the increment in thickness upon chitosan adsorption? It seems rather larger for a layer-by-layer deposition.

Answer: We agree with the reviewer that the increase in particle size from LNPs (97nm) after the chitosan adsorption (chi-LNPs (190nm)) is larger than expected for a simple layer-by-layer deposition. However, it is important to note that in the present study, we aimed to simplify the LNPs production process and did not use dialysis as a purification step for the solvent removal, therefore the presence of some low molecular weight impurities bearing carboxylic acid groups could affect the thickness by precipitating with chitosan on the lignin surface with the consequent increase on the thickness of the particles after adsorption process. We have revised discussion on this aspect as follows: **"In the present study, we aimed to develop a material-efficient process and in order to simplify the LNPs production process we did not use dialysis as a purification step but instead evaporation for the solvent removal. Therefore, the larger than expected increase in the particle diameter after the chitosan adsorption could be related to the co-precipitation of chitosan and small lignin compounds bearing carboxylic acid groups on the LNPs surface".** Please, see page 2, right column lines 1-5 from top of the red market manuscript)

2) What is the partition of the ligand between the oil and water phase? The author should discuss the need of some Me₆TREN in the water phase to stabilize the Cu²⁺

Answer: We really appreciate the reviewer attention on the mechanism of SET-LRP in Pickering emulsions and especially on the partition of the reagents during the polymerization process. In fact, we received inspiration to our system from the recently developed “programmed” biphasic SET-LRP. In this system the partition of the reagents and the migration of Cu(I)-Me₆-TREN complex to the water phase after the activation step is crucial to control the polymerization process. In this sense, we have revised discussion on this aspect as follows: “It is important to note that in our system, activation, disproportionation and deactivation steps take place in different compartments *via* partitioning of the ligand and copper active species as has been recently postulated for “programmed” biphasic SET-LRP systems.^[37-39] In this sense, the activation step proceeds in the organic phase (oil phase) *via* out sphere single electron transfer process (OSET), where Cu(0) acts as an electron donor to promote the heterolytic cleavage of the alkyl halide bond to generate the active radicals.^[40] Afterwards, the generated Cu(I)-Ligand complex formed during the activation step migrate to the aqueous phase where the crucial disproportionation step takes place, giving rise to the Cu(II)-ligand complex and *in situ* nascent Cu(0). In this context, the deactivation step is postulated to take place in the interphase between aqueous and oil phase, *via* a reverse OSET process mediated by Cu(II)-Ligand complex.^[37] Therefore, the surface area of the polymeric microdroplets and an efficient partition of the ligand between aqueous and organic (oil) phases are crucial aspects in our system to provide an efficient deactivation step and to provide control over the polymerization.” Please, see page 3, left column lines 22-42 from top of the red market manuscript). We also included important references (37, 38, 39 and 40), related to the mechanism of SET-LRP in biphasic mixtures. We believe that the revised text addresses the review concerns.

3) “linearity between indicating an equal propagation of the growing polymeric chains, the principal feature of a living polymerization” The principal feature of a living polymerization is the ability to re-initiate and chain extend. The linear plot of Ln(M/M₀) vs t is a feature of control in the degree of polymerization. The authors should rephrase their sentence.

Answer: Thank you very much again for this constructive comment and carefully reading our manuscript. We agree with you that one of the main features of living radical polymerization process is the ability to reinitiate a polymeric chain to get access to more complex architectures. We have revised discussion on this aspect as follows: “indicating an equal and constant propagation of the growing polymeric chains, which essentially confirms a lower frequency of the termination events (i.e. recombination of growing polymeric chains). These results, together with a high retention of chain end functionality, and the ability to reinitiate the polymeric chain (*vide infra*), prove unequivocally the living radical polymerization behavior for our process.” Please, see page 3 left column, lines 15-22 from bottom of the red marked manuscript.

4) Figure 3 reports the comparison between GOx-chit-CLPS and chi-CLPs. It is reassuring to see that the enzyme was capable of degassing and protecting the system (compared to N₂ degassing) and prevented any induction period. I am confused however, why in the plot of M_n (GPC) vs M_n (theo) all the kinetics lay on the same curve. I would expect that the presence of oxygen would cause termination and a concomitantly lower initiation efficiency. The manuscript would benefit from the calculation of the initiation efficiency and chain-end functionality in this comparison.

Answer: We really appreciate the carefully reading of the reviewer of our manuscript and also appreciate pointing out this discrepancy. We were already concerned about this honest mistake during the review process and realized that we had plotted the incorrect data for the reaction of chi-LNPs in the presence of air (green squares, Figure 3E). We had placed instead data from a duplication control reaction using GOx-chi-LNPs. We have replaced the figure with the correct data (See new figure 3E, in red market version of the manuscript). As can be seen from the new version of the figure, a clear deviation on molecular weight values for chi-LNPs in the presence of air (squares) can be observed especially at the beginning of the reaction, indicating a termination process by the coupling of oligomeric growing chains with the dissolved oxygen in the reaction, as correctly stated by the reviewer. Additionally we also included in the figure the calculation of the initiator efficiency (I_{eff}) as the reviewer suggested which also proves a lower I_{eff} for chi-LNPs in the presence of air in comparison to GOx-chi-LNPs or chi-LNPs (under N_2). We also revised the Results section as follows: "These results confirm the potential of lignin nanoparticles as functional emulsifiers for SET-LRP. However, it is important to note that in the case of chi-LNPs without applying any degassing procedure, a clear deviation from the M_n (GPC) values in comparison to the theoretical ones M_n (th), can be appreciated, especially at the early stages of the reaction (Figure 3e, squares). This fact, together with a decrease in the initiator efficiency (I_{eff} , Figure 3e), is associated to the consumption of the dissolved oxygen at the beginning of the polymerization process by the growing oligomers radicals." Please, see page 3 right column, lines 1-11 from bottom of the red marked manuscript.

We also incorporated the determination of the chain end functionality for the [PMA (M/I) = 50] sample obtained using chi-LNPs in the presence of air, following the reviewer's suggestion. The results are incorporated in the new Figure S11 of the SI, which essentially indicates a lower chain end functionality (85% in comparison to 96% for GOx-chi-LNPs), confirming some termination processes during the polymerization. We also revised the Results section as follows: "For comparative propose, PMA (M/I =50) using chi-LNPs and without the application of any deoxygenating protocols was also synthesized. In this case, the chain end analysis of the sample revealed a lower functionality (85%) (Figure S11), in comparison to the above mentioned, which essentially confirms the presence at some extent of bimolecular termination (i.e. by the coupling of growing oligomeric radicals with dissolved oxygen)." Please see page 4 right column, lines 5-13 from bottom of the red marked manuscript.

5) What do the author mean with "quasi real AB block copolymers"?

Answer: Although we determined that our polymers exhibit a high chain end functionality by 1H NMR and can be used successfully to obtain block copolymers by simple chain extension experiments, we also assume that the presence of very minor termination events can be a possibility. In this context, we called "quasi real AB block copolymers" giving a room to the possibility of a very minor extent of unreactive dead polymeric chains. However, we agree with the reviewer that such a statement can be misunderstood, since this term is also used to refer to block copolymers consisting of conventional and supramolecular blocks, where the conventional block interacts with a supramolecular monomer acting as a bridging linker. Therefore, based on our findings (almost quantitative chain end functionality and "clean chain extension experiments" we have revised the Results section in this aspect as follows: "The successful formation of well-defined block copolymers". Please see, page 4 left column, line 2 from bottom of the red marked manuscript.

6) The direct use of the stabilized polymer emulsion is interesting for many applications. Have the authors check if there is any remaining Cu in the polymer phase? And how much is it?

Answer: Thank you for arising this very important point. We agree with the reviewer that in general the amount of Cu in Copper-mediated living radical polymerization techniques is important, especially for determined potential application in biological fields. For a typical polymerization

process, with a theoretical DP of 200, initially we chose a mole ratio of Cu(0) to initiator of 0.3. Therefore, the reaction mixture only contains 1.19 mg of copper, which corresponds to a maximum concentration of 119 ppm. It should be noted that during the optimization of the reaction conditions the amount of copper could be reduced to a half of this (0.15 molar ratio, 0.60 mg, and 60 ppm) without affecting the control over the polymerization and with the only drawback of reducing the polymerization rate constant (two times). Nevertheless, the values used in our work are substantially lower in comparison to other reports in the literature, using around 600 ppm of copper in a combination of Cu(II) and Cu(0)^[1], or 250 ppm of Cu(0) in a miniemulsion process,^[2] which could lead to an important contamination of the resulting aqueous solution.

Following the suggestion of the reviewer, and in lack of access to quantitative elemental analysis (e.g. by XPS or ICP-MS), we analyzed the amount of copper present on a PBMA-lignin composite by energy-dispersive X-ray (EDX) spectroscopy. The amount of Cu(0) was close to the detection limit (please see Figure R1), which only indicates that the amount of copper is less than 100 ppm (<0.1% - 0.3 % atomic level) according to the lower detection level of our instrument. However, it is important to state that the presence of copper traces in the final composite could be expected, since lignin has affinity to adsorb copper (15 mg/g).^[3] On the other hand, the GOX-chitosan-lignin hybrid particles displayed a positive net charge, which should be repulse adsorption of copper cations. This means that without any additional external treatment to remove lignin (i.e. basic treatment) we can speculate that an amount of less than 50% respect to the initial amount of copper could remain on the final material. As we stated in the manuscript, we believe that the system presented in this work is robust enough to be transferred to other RDRP using a lower amount of copper as catalyst if the final application requires it. For instance, good alternatives would be activators regenerated by electron transfer ARGET-ATRP and initiators for continuous activator regeneration ICAR-ATRP, which typically use concentrations below to 10 ppm of copper, or even RAFT without the presence of copper as catalyst.

Figure R1. EDX analysis of PBMA-GOx-chi-LNPs composite after enzyme-degassed SET-LRP-mediated Pickering emulsion process. (a) Specific surface area used for the analysis. (b) Composition of the sample in wt% and (c) EDX spectra of PBMA-GOx-chi-LNPs using 10 kV as accelerated voltage.

^[1]Konolewicz, D., Kryś, P., Gójs, J. R., Mendonça, P. V., Zhong, M., Wang, Y., Gennaro, A., Isse, A. A., Fantin, M. & Matyjaszewski, K. Aqueous RDRP in the presence of Cu(0): The exceptional activity of Cu(I) confirms the SARA ATRP mechanism. *Macromolecules* **47**, 560-570 (2014).

^[2] Elsen, A. M., Burdyńska, J., Park, S. & Matyjaszewski, K. Active ligand for low ppm miniemulsion atom transfer radical polymerization. *Macromolecules* **47**, 7356-7363 (2012).

^[3] Ge, Y. & Li, Z. Application of lignin and its derivatives in adsorption of heavy metals ions in water: a review. *ACS Sustainable Chem. Eng.* **6**, 7181-7192 (2018).

We also revised the Results section as follows: "It is also interesting to note that the unpurified polymer composite contained less than 0.2% of copper when analyzed by SEM-EDX, while showing 2.7 atom-% of bromine linked to the active chain-ends (Figure S16). However, it is important to state that the presence of copper traces in the final composite could be expected, since lignin has affinity to adsorb copper (15 mg/g).^[48] On the other hand, the GOX-chitosan-lignin hybrid particles displayed a positive net charge, which should be repulse adsorption of copper cations." Please see page 5 right column, lines 15-24 from top of the red marked manuscript.

In summary, this is an excellent contribution of Moreno and Sipponen, which allows for the polymerization of hydrophobic polymers in water without the need of any degassing and even protect the polymerization from oxygen coming into contact during the polymerization. This is an excellent way to achieve polymers with controlled distribution of molecular weight and high chain end functionality in very facile manner. I am convinced that the findings presented here will be of great interest for a broad readership and I recommend the acceptance of this manuscript.

Thank you again for your very positive feedback and carefully reviewing of our work, which allowed us to improve the current version of the manuscript. We hope that revised text and responses address your concerns.

César Rodríguez-Emmenegger

Reviewer #2 (Remarks to the Author):

This is an excellent manuscript that will impact the field of living radical polymerization and therefore I recommend publication after minor revisions. The following minor revisions must be considered by the authors: (1) In reference 10 the name of the first author must be corrected to S. Fleischmann and of the third author to V. Percec. The following additional citations on the topic of reference 10 could be cited: *JPS Polym Chem* 2010, 48, 2243; *JPS Polym Chem* 2013, 51, 3110 and *JPS Polym Chem* 2011, 49, 4756. (2) In Figure 4a,b and S5b and S6a,b,c replace Mn/Mw with Mw/Mn. After these minor revisions this excellent manuscript is recommended for publication.

Thank you very much for reviewing our manuscript, the very positive comments about our work and recommending it for publication after minor revisions.

Answer (1): Thank you very much for these very important citations. We have revised our version and included new references 13, 14 and 15 as well as corrected the spelling of the author's name in

reference 11. We also incorporated new references 37, 38, 39 and 40 related the mechanism behind the biphasic SET-LRP systems.

Answer (2): Thank you very much for noting this honest mistake. We have revised and corrected the figures 4 and S5+S6.

REVIEWERS' COMMENTS

Reviewer #1 (Remarks to the Author):

The authors have carefully and correctly addressed all the questions I had. I believe it is an excellent paper and recommend its publication.

Manuscript number: NCOMMS-20-29651

Manuscript type: Research article

Title: Biocatalytic Hybrid Nanoparticles for Degassing and Stabilization of Single Electron Transfer-Living Radical Polymerization in Pickering emulsions

Authors: Adrian Moreno, Mika H. Sipponen

Correspondence to: mika.sipponen@mmk.su.se

REVIEWERS' COMMENTS

Reviewer #1 (Remarks to the Author):

The authors have carefully and correctly addressed all the questions I had. I believe it is an excellent paper and recommend its publication.

Responses to Reviewer's comments:

We would like to thank the expert reviewers for their critical comments on our manuscript and the positive feedback about our work.